# VisionAD, a software package of performant anomaly detection algorithms, and Proportion Localised, an interpretable metric

**Alexander D.J. Taylor**                                   *adjt20@bath.ac.uk*
*ART-AI*
*Department of Computer Science*
*University of Bath*

**Jonathan James Morrison**                       *jonathan.morrison2@rolls-royce.com*
*Defence Operations*
*Rolls-Royce*

**Phillip Tregidgo**                                   *phil.tregidgo@rolls-royce.com*
*Defence Operations*
*Rolls-Royce*

**Neill D.F. Campbell**                                   *nc537@bath.ac.uk*
*ART-AI*
*Department of Computer Science*
*University of Bath*

**Reviewed on OpenReview:** *https://openreview.net/forum?id=o5kYH7bNe3*

## Abstract

We release VisionAD, an anomaly detection library in the domain of images. The library forms the largest and most performant collection of such algorithms to date. Each algorithm is written through a standardised API, for ease of use. The library has a focus on fair benchmarking intended to mitigate the issue of cherry-picked results. It enables rapid experimentation and straightforward integration of new algorithms. In addition, we propose a new metric, Proportion Localised (PL). This reports the proportion of anomalies that are sufficiently localised via classifying each discrete anomaly as localised or not. The metric is far more intuitive as it has a real physical relation, meaning it is attractive to industry-based professionals. We also release the VisionADIndustrial (VADI) benchmark, a thorough benchmarking of the top anomaly detection algorithms. This benchmark calculates the mean across the pooled classes of the MVTec and VisA datasets. We are committed to hosting an updated version of this leaderboard online, and encourage researchers to add, tweak and improve algorithms to climb this leaderboard. VisionAD code is found at `https://github.com/alext1995/VisionAD`, and Proportion Localised code is found at `https://github.com/alext1995/proportion_localised`.

## 1 Introduction

In the real world many classification problems do not contain sufficient data of various classes. Therefore the supervised training of deep learning models presents a challenge. Due to this, the field of anomaly detection has arisen. Anomaly detection refers to the process of training a model using only nominal samples, where the model is intended to differentiate nominal and anomalous samples during inference. The de facto dataset in the field of visual anomaly detection is MVTec, from Bergmann et al. (2019). MVTec contains over 5000

images across 15 classes of various industrial and household objects. The field of anomaly detection currently enjoys growing attention, with a number of algorithms recently released (Yu et al., 2021; Roth et al., 2022; Gudovskiy et al., 2022; Tsai et al., 2022; Wan et al., 2022b;a; Cao et al., 2023; Lee et al., 2022; Kim et al., 2022; Bae et al., 2023; Deng & Li, 2022; Rudolph et al., 2022). The VisA dataset, by Zou et al. (2022), contains 12 classes of household and industrial objects, and is more challenging than the MVTec dataset. It has not experienced the same attention as MVTec, but we propose that it has equal value due to its challenging nature and varied classes.

The PapersWithCode (PWC) leaderboard, hosted by MetaAI (2022), can be used to judge the performance of these algorithms for the MVTec dataset. However, the results from the publications on the leaderboard are self-reported, and the leaderboard is editable by anyone. This means that the positions on the leaderboard may not be representative of the relative performance of the algorithms. If researchers wish to experiment with different algorithms, they currently need to download many different GitHub repositories, which contain different data-loading code, metric systems, styles and environments. Despite the best intentions of researchers to make their code friendly to use, this overhead is significant.

The dominant metrics in the field of anomaly detection have a number of disadvantages. Image-wise AUC (I-AUC) works at the image level, and does not measure localisation of anomalies. Pixel-wise AUC (P-AUC) and Area Under Per-Region Overlap (AUPRO) contain noise that stems from ambiguous pixel labelling, produce values in a small range, and lack a physical relation.

In light of the above issues, we make the following contributions:

1. We release a comprehensive library of anomaly detection algorithms, VisionAD. All algorithms are standardised through an API. The algorithms share the same data-loading and evaluation code, meaning that fair benchmarking is possible. The library has a focus on ease of use, with a simple system for researchers to implement their own algorithms.

2. We release a new metric, Proportion Localised (PL), for anomaly detection. This classifies each anomaly as discretely localised or not, and reports the proportion of the anomalies 'localised'. This is interpretable, and negates the noise associated with the labelling of ambiguous pixels. A standalone Python "PIP" package for this metric is published.

3. We undertake a benchmarking of the currently implemented algorithms on the MVTec and VisA datasets. We refer to this as the VisionADIndustrial benchmark. An online leaderboard will be updated as and when new algorithms are released.

## 1.1 Comparison of VisionAD and Anomalib

The most similar library currently available is Anomalib, from Akcay et al. (2022). As of writing, Anomalib contains 14 anomaly detection algorithms. The library contains shared code for metric and data-handling. Algorithms are also available to export in OpenVINO and ONNX, and the library comes with a number of tools such as Lightning, Gradio inference, hyperparameter sweeping, and integrated web UIs. However, these extensive features of Anomalib may result in costly implementation of algorithms, as algorithms must comply with all of these features. This may also impede rapid experimentation.

VisionAD is designed to allow researchers to add their own algorithms very easily if they wish. This is intended to encourage VisionAD to grow fast. The user-friendly focus of VisionAD is designed to enable quick experimentation. For instance, a researcher can load the provided IPython file and immediately start the modelling process, as data-loading and evaluation are handled. Implementation of the code into the library can be achieved by copying the experimental code, as the IPython file ensures the API structure is followed.

The algorithms of Anomalib are overall less performant and older than those of VisionAD, according to the PWC leaderboard (MetaAI, 2022). VisionAD also contains more algorithms, meaning it is the largest library of its kind, and it is hoped it will grow further as new publications are released.

The authors do not intend to discredit Anomalib, which has enjoyed a lot of community attention and is a positive addition to the field. The authors propose that VisionAD can provide value to the community by overcoming the shortfalls of Anomalib. We propose that VisionAD and Anomalib have different strengths and complement each other. Anomalib has rich features for analysis and deployment, whilst VisionAD provides a lightweight alternative with a focus on rapid development and benchmarking.

### 1.2 Algorithms

Many different classes of algorithms exist in the anomaly detection space. Transfer learning makes use of a backbone network pre-trained on ImageNet (Deng et al., 2009), and is found in almost all recent anomaly detection algorithms. The most performant algorithms tend to make use of one or more of the concepts described below. Note the descriptions below are not exhaustive, and we refer the reader to review works by Pang et al. (2020) and Thudumu et al. (2020) for more information.

Distillation algorithms make use of the concept that during inference, the difference between the predictions of a teacher model and a student model should highlight anomalous regions. Some algorithms of this class are CDO (Cao et al., 2023), RSTDN (Yamada et al., 2022), AST (Rudolph et al., 2022), and EfficientAD (Batzner et al., 2024).

Normalising flows map data of an arbitrary distribution to a normal distribution whilst conserving volume, using learnable bi-directional transformations. Usually, features created via a pre-trained model are mapped via normalising flows to a normal distribution. In the case of anomalous inputs, the normalising flow function should fail to create a normal distribution, as the function has not seen the anomalous data before. Examples of algorithms that fall into this category are Fastflow (Yu et al., 2021), CFlow (Gudovskiy et al., 2022), MSFlow (Zhou et al., 2024), and Fastflow+Altub (Kim et al., 2022).

Memory bank algorithms extract and process features from the training data, then filter or reduce these features before saving them into a memory bank. During inference, test features are processed and compared to the features in the memory bank via some distance or nearest-neighbour measure. A larger distance implies that a feature belongs to an anomalous region. Algorithms of this class are PatchCore (Roth et al., 2022), PNI (Bae et al., 2023), MemSeg (Yang et al., 2022), and CFA (Lee et al., 2022).

Synthetic anomalies can be used to either train a model using traditional supervision or to refine the outputs using supervision. These anomalies can be entirely synthetic using a mathematical formula such as Perlin noise, or can be constructed using examples from other datasets. Algorithms which make use of this concept are PNI (Bae et al., 2023), MemSeg (Yang et al., 2022), CDO (Cao et al., 2023), and SimpleNet (Liu et al., 2023b).

Generative algorithms learn to recreate the input, with the intention that during inference, the model will fail to recreate an anomalous region, therefore highlighting its existence. An algorithm which makes use of this concept is PPDM (Liu et al., 2023a).

Other algorithms use novel concepts altogether. For example PFM (Wan et al., 2022a) and PEFM (Wan et al., 2022b) train different models to recreate the same output when given an image. The difference in output should indicate anomalous regions.

## 2 VisionAD

We release VisionAD, a library comprising of many of the best anomaly detection algorithms, all written such that they communicate with the library using a standardised API. Figure 1 shows the architecture. The wrapper is responsible for loading the configuration file, calling the data-loading code, calling the algorithm through the various standardised API calls, evaluating the performance using the specified metrics, and logging the results. A user guide can be found in Appendix A.

## 2.1 Implemented algorithms

The algorithms currently implemented are: FastFlow (Yu et al., 2021), PatchCore (Roth et al., 2022), SimpleNet (Liu et al., 2023b), PPDM (Liu et al., 2023a), MSFlow (Zhou et al., 2024), PFM (Wan et al., 2022a), PEFM (Wan et al., 2022b), CFlow (Gudovskiy et al., 2022), MSPBA (Tsai et al., 2022), MemSeg (Yang et al., 2022), FastFlow+AltUB (Kim et al., 2022), CDO (Cao et al., 2023), CFA (Lee et al., 2022), Reverse Distillation (Deng & Li, 2022), EfficientAD (Batzner et al., 2024), and AST (Rudolph et al., 2022). As of March 2024, these comprise the top 15 algorithms with code available on the following columns of the MVTec leaderboard, I-AUC, P-AUC, and AUPRO. We choose not to implement algorithms from scratch, due to the risk of introducing errors and unfair representation. AltUB is the one exception to this rule. It was implemented entirely from scratch due to its simplicity.

Algorithms were reformatted from their existing structure to fit the standardised structure necessary for VisionAD. Utmost care was taken to ensure the algorithm formatting was done without errors. Implementations were based on official code where possible, and unofficial code where necessary. MSPBA, Fastflow and MemSeg were based on unofficial code. We give due credit to the authors of these unofficial repositories in the code base. Whilst we cannot guarantee implementations are perfect representations, algorithms were tested against the bottle and carpet class of MVTec to ensure the same results were achieved by the repository and reformatted code.

The authors are committed to adding more algorithms as they are published, and encourage researchers to integrate their own algorithms via pull requests.

Certain aspects of the code inherit the licences of the previous code. Therefore care must be taken regarding use of the library. Due to this, the authors recommend academic and non-commercial use.

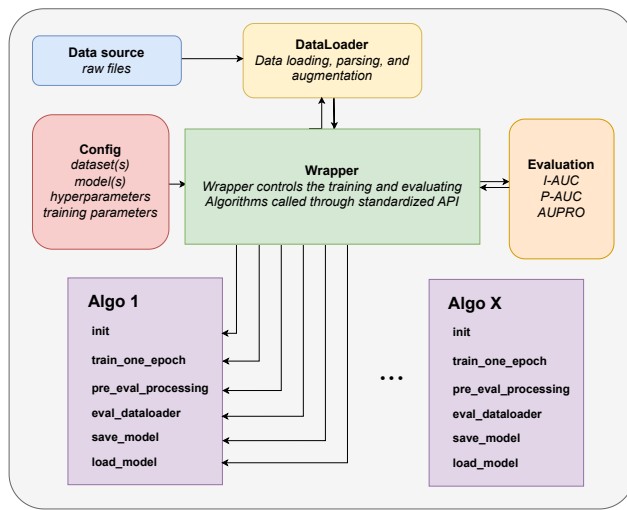

Figure 1: Flow diagram showing the architecture of the VisionAD library.

## 2.2 Features and use

Adding algorithms is intended to be as easy as possible, as the researcher needs to implement six methods of a Python class for full integration with the library. VisionAD contains efficient implementations of the standard metrics (I-AUC, P-AUC, AUPRO), alongside the introduced PL, which is discussed below. Implementing a new metric is also simple. Due to the ease of writing complex run configurations in Python, Python configuration files control the behavior of each run. Only minimal command line arguments are used: the path to the config file, the GPU device, and an optional run description. For development purposes, an IPython notebook is provided with an algorithm that makes random predictions, allowing researchers to experiment in an interactive environment. A more detailed description of the above features can be found in Appendix A, and a guide to getting started quickly is available in the readme.md file.

## 3 Proportion Localised (PL) metric

We propose a new anomaly detection metric, Proportion Localised (PL). The motivation of PL is to provide a metric that summarises the performance of an algorithm by evaluating whether it sufficiently highlights the approximate areas of the anomalies, as opposed to whether the algorithm precisely segments each pixel. We propose that this is more suitable for industrial applications in which an algorithm functions correctly if it 'locates' the anomaly, as opposed to whether it replicates the exact shape at the pixel level. We also

state the desire for a metric which reports a number with a real physical relation, so that it can be readily understood by non-machine learning professionals.

In summary, for each anomaly, PL calculates an IoU score between each prediction heatmap and the rotated bounding box for the anomaly. To obtain the binary prediction heatmap, the continuous algorithm output needs to be put through a prediction threshold. Here a sweep is done across many prediction thresholds. For each resultant binary prediction heatmap, the proportion of anomalies with a prediction and ground truth IoU greater than the IoU limit is calculated, and the highest value amongst this array is reported as the final value. We use 0.3 as the IoU limit (we explain this choice below). Therefore PL reports the proportion of anomalies which are sufficiently found at the best threshold. The calculation is described in detail in Section 3.4. Looping through different prediction thresholds and taking the best score means the best prediction threshold is used. This means that like P-AUC and AUPRO, the algorithm is independent of the prediction threshold.

The minimum area rotated bounding box is drawn around each anomaly, and these bounding boxes act as the ground truth. This is intended to remove the noise of pixel by pixel matching. Due to this, we deliberately state that the metric evaluates anomaly *localisation*, as opposed to anomaly detection (image-wise), or anomaly segmentation (pixel-wise). Many experts can not agree on the classification of bordering pixels of anomalies, and due to this, we believe that an algorithm should be judged via whether it sufficiently highlights the region of the anomaly, as opposed to classifying each pixel exactly. An algorithm cannot classify every pixel correctly due to this ambiguity, and when evaluated using the traditional P-AUC and AUPRO, cannot achieve perfection. Furthermore, we emphasise that for many industrial applications, an anomaly detection algorithm functions as required if it highlights the approximate location and size of the anomaly. Unlike AUC and AUPRO, PL can report perfection if the algorithm produces an IoU greater than the required IoU limit (0.3) for every anomaly.

## 3.1 Weaknesses of current metrics

Table 1 summarises the drawbacks of the current metrics and the advantages of PL. I-AUC fails to properly measure images with more than one anomaly, and fails to measure any degree of anomaly localisation or segmentation. The pixel-wise metrics, P-AUC and AUPRO, amplify any ambiguity in the labelling of the pixels. Experts disagree on the individual classification of certain pixels, so therefore it is an impossible task for an anomaly detection algorithm to achieve perfect scores on these metrics. In addition to this, anomaly detection algorithms tend to output prediction heatmaps at a lower resolution than the input image, meaning it is impossible for most algorithms to extract the exact shape of an irregular anomaly. This all contributes to noise in the output of these metrics. We argue that anomaly detection algorithms function as intended if they highlight the approximate area of the anomaly.

The pixel-wise metrics also suffer from vastly optimistic scores due to the large imbalance of anomalous and regular pixels. The P-AUC metric has a low range of output. On an imbalanced dataset, a bad score may be 0.9 and a good score may be 0.99. This is undesirable, it is preferable if a bad score is 0 and a good score is 1. The sIoU metric (Chan et al., 2021) is innovative, and recognises the drawbacks of P-AUC and AUPRO by functioning at the anomaly level. However it does have a number of downsides. It is slow to run, due to the looping where the predictions have to be separated into non-overlapping regions for many different thresholds. In addition, the multiple counting of false positive prediction area for different anomalies in a single image is over-penalisation.

| P-AUC/AUPRO
drawbacks | PL
advantages |
|---|---|
| Pixel-level matching contains ambiguity and noise | Less effected by pixel-level noise through BB use |
| Reports an abstract number (no physical relation) | Reports a number with a real physical relation |
| Small output range, usually predicts between 0.8 and 1.0 | Larger output range, can predict between 0.0 and 1.0 |
| Lack of understandability for non-ML professionals | Readily understandable by non-ML professionals |

Table 1: Drawbacks of P-AUC and AUPRO, and advantages of PL.

Finally, none of these metrics are readily understood by users without machine learning expertise. They do not report a number with a physical relation.

## 3.2 Strengths of the introduced PL

PL directly reports the proportion of anomalies localised. Engineers, inspectors and doctors are likely interested in what anomalies have been 'localised' (found) by an algorithm, where localised is a binary descriptor, and is *true* if the given algorithm has highlighted enough of the anomaly, and *false* if not.

A major advantage of PL is that it is readily understandable by non-ML professionals. One could understand that PLs of 0.1, 0.5, and 0.99 mean that 10%, 50%, and 99% of anomalies are sufficiently located. We propose that these numbers could be used to make an informed decision in an industrial and medical setting. We emphasise that the friendliness of this metric to industry professions is a major advantage.

PL uses rotated bounding box labels, which negates the issue of noise in pixel-wise matching, as algorithms are not judged on whether

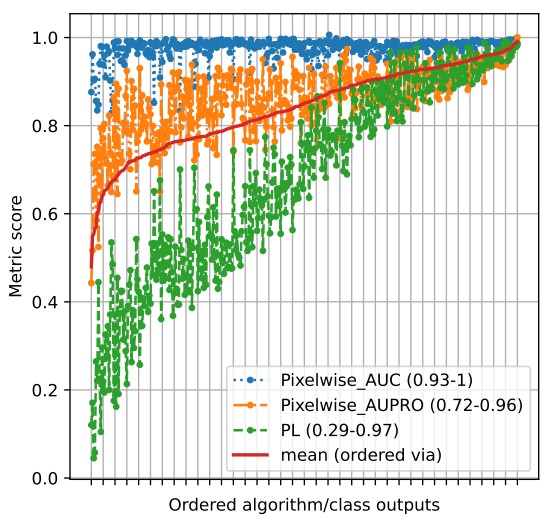

Figure 2: Comparing the behaviour of PL to the traditional metrics, (5th - 95th percentile)

than locate the exact pixels. Instead they are judged on whether they sufficiently locate the bounding box area of the anomaly. This allows algorithms to output heatmaps at a lower resolution to the image without penalisation (many algorithms do this). It also removes the requirement that algorithms must replicate the exact shape of an anomaly. This brings anomaly detection in accordance with the object detection field, which uses IoU to measure overlap to discretely classify whether an object is found or not (mAP, AP families) (Lin et al., 2014; Redmon et al., 2015; Girshick et al., 2014).

Finally PL uses a larger range of outputs in comparison to the other common metrics. Behaviour of the PL metric is compared to P-AUC and AUPRO in Figure 2.

## 3.3 Behaviour of PL

Figure 2 shows the behaviour of the PL metric compared to the P-AUC and P-AUPRO metric. Each vertical set of three points represents the P-AUC, P-AUPRO, and PL outputs of a specific algorithm and dataset class combination. These are ordered based on their mean value across these metrics. The classes used are from MVTec and VisA. The relative value of each metric is of importance here, not the performance of the algorithms. Firstly we note that P-AUC predicts in a much smaller range of values, with most predictions greater than 0.95. AUPRO produces a greater range of values, but the values are still generally between 0.8 and 0.95 (with a small number of algorithm class combinations greater than this). Meanwhile PL predicts across a much greater range (0.1-0.98), and has much more linear behaviour.

## 3.4 Calculation of PL

Algorithm 1 outlines the calculation. For each anomaly ground truth and corresponding prediction, the algorithm loops through 25 linearly spaced prediction heatmap thresholds, and creates a 2d array of shape (no.anomalies, 25). For each threshold, the proportion of anomalies with an IoU greater than the IoU limit (0.3) is calculated. The final score is the maximum of these.

---

**Algorithm 1:** Calculation of Proportion Localised. Pred_set and GT_set are the groups of prediction heatmaps and ground truth images respectively. Details of the sub functions can be found in Appendix C

---

```
1  function Calculate_PL(pred_set, GT_set)
2      IoUs ← []
3      for prediction_heatmap, GT_image in pred_set, GT_set do
4          BBs ← ExtractRotatedBoundingBoxes(GT_image)
5          BBs ← ScaleBBs(BBs)
6          BBs ← MergeOverlappingBBs(BBs)
7          filters ← CreateNearestBBFilters(BBs)
8          for single_BB, filter in BBs, filters do
9              comb_filter ← LogicalOr(single_BB, filter)
10             pred_filtered ← comb_filter * prediction_heatmap
11             IoUs_per_threshold ← []
12             for threshold in thresholds do
13                 binary_pred ← pred_filtered > threshold
14                 IoU ← CalcIoU(binary_pred, single_BB)
15                 IoUs_per_threshold ← append(IoU)
16             IoUs ← append(IoUs_per_threshold)
17     IoUs ← array(IoUs)
18     ret ← IoUs > IoU_limit
19     ret ← MeanAxis0(ret)
20     ret ← Max(ret)
21     return ret
```

There are a few nuances which make the process work. Firstly, if labels are pixel-wise, they need to be converted to freely-rotated bounding boxes. This is easily achieved by creating the minimum area bounding box over every non-overlapping ground truth region. The bounding boxes are freely rotated to best match the shapes of the anomalies, and to ensure the metric is independent of image rotation.

It was found that during inference very small anomalies cause activity on the heatmap of an area much larger than the anomaly label. This is because most anomaly detection algorithms output prediction heatmaps at resolutions lower than the input image, *e.g.* $32 \times 32$ in comparison to $256 \times 256$.

If an anomaly is very thin, it is near-impossible for the algorithm to achieve the desirable IoU, even if the algorithm prediction is sufficiently centered on the anomaly. We do not believe this should be penalised, as in this case the algorithm has still located the anomaly. Therefore in the PL calculation, the bounding boxes are enforced to have a minimum width and height in relation to the respective image dimension. A value of $1/8^{\text{th}}$ (12.5%) was found to work well. This allows coarse prediction heatmaps to be judged fairly. This minimum dimension enforcement of $1/8^{\text{th}}$ of the image dimension is part of the introduced PL metric. The scaling is shown graphically via the difference between the first and second rows of the examples in Figure 3. This scaling results in a small boost to the PL scores, which we believe is justified, as it prevents penalising relatively big predictions centered on small anomalies.

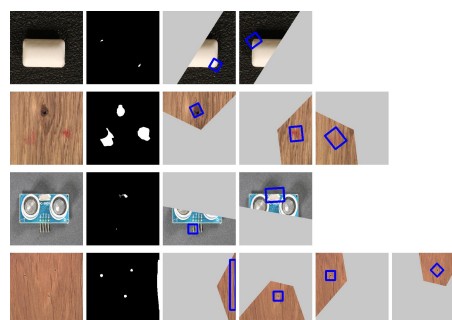

Figure 3: Demonstration of splitting of images with more than one anomaly. Columns left to right: input image, raw ground truth, subsequent splits (blue bounding box mask, grey ignored area).

The metrics handle an arbitrary number of anomalies in one image by splitting the prediction heatmap at the boundaries between the centers of each anomaly (each pixel of the heatmap is designated to its nearest anomaly). For a given anomaly $x$, every pixel which is not closest to the anomaly center point, or inside the bounding box, is zeroed before the IoU calculation. The process is shown graphically in Figure 3 via the

$3^{\text{rd}}$, $4^{\text{th}}$, and $5^{\text{th}}$ rows. In the subsequent steps, each anomaly and heatmap region is handled independently, meaning the metric functions at the anomaly level, as opposed to the image or pixel level

With the process explained above, a group of small and close anomalies would result in a large number of overlapping bounding boxes, which is undesirable. If there is significant overlap in the scaled bounding boxes, the anomalies should be merged and treated as a single anomaly. Bounding boxes which overlap by more than $1/3^{\text{rd}}$ with another are merged. The underlying pixel-wise ground truths are merged, and a new rotated bounding box is drawn and scaled. This process is shown graphically in Figure 4, and explained in more detail in Appendix C.

PL requires the choice of an IoU limit. Only anomalies with a prediction and ground truth IoU greater than this limit are classified as located. We use and recommend an IoU limit of 0.3, although a researcher can use the value they see the most fit for their problem. We justify the choice of 0.3 in two ways. Firstly, we take precedence from the AUPRO calculation, from Bergmann et al. (2019), which calculates the area under the curve up to a false positive rate of 0.3. Secondly, we use Figure 5 to undertake a visual analysis of predictions and bounding box calculations across the MVTec and VisA datasets. We wish to choose a IoU limit which excludes unsatisfactory overlap between the prediction and label, whilst including satisfactory overlap. For simplicity, we wish to use a value which is a multiple of 0.05. We believe 0.3 is the best value which separates satisfactory and non-satisfactory overlaps. Part way of the third row of Figure 5 and onwards shows satisfactory overlap between the prediction and bounding box, whilst the images before it do not. The value of 0.3 may be perceived as low, but this allows some inevitable mismatch be-

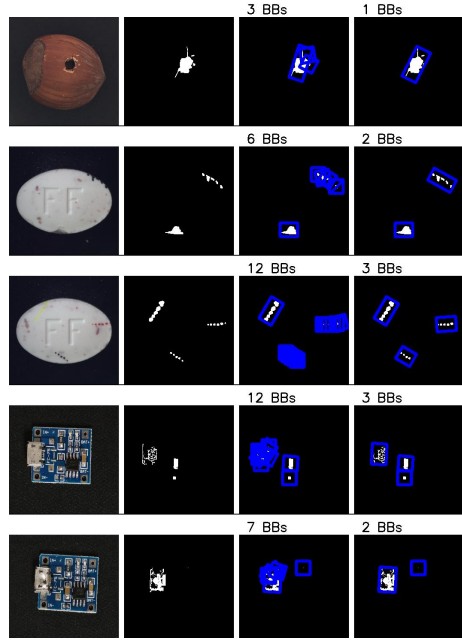

Figure 4: Demonstration of merging overlapping anomalies. Columns left to right: input image, raw ground truth, bounding boxes around non-overlapping regions (no merging), bounding boxes after merging.

tween the bounding box target and the prediction. Note that some predictions produce what we perceive to be good overlap, but still have IoU values between 0.4 and 0.6. Whilst the published PL uses 0.3 as the limit, the researcher can choose any value, and this can be represented by PLxx, i.e. PL50 for 0.5 limit, and PL25 for 0.25 limit.

The metric does not require the use of any regular testing images. The reader may assume that the PL metric does not penalise false positive predictions. *PL does penalise false positive predictions.* As the anomalous images still contain a significant amount of non-anomalous regions, if an algorithm were to predict non-anomalous regions as anomalous, the IoU of the closest anomaly would be reduced such that it would not meet the IoU limit, and would not be classified as localised by PL, therefore reducing the final score.

A Python package to calculate PL is also released to allow researchers to quickly integrate these metrics into their code. The metric is coded efficiently. It takes approximately the same amount of time to calculate PL as it takes to calculate P-AUC or AUPRO.

## 4 VisionADIndustrial (VADI) benchmark

### 4.1 Results

We introduce the VisionADIndustrial benchmark. For the respective benchmark, this is the mean value of the pooled class scores of the MVTec and VisA datasets. Effectively we combine the datasets. The scores for each respective class in both datasets are added and then we divide by the total number of classes (27). This ensures each class is weighted equally. We do not calculate the individual mean of the MVTec and VisA datasets, and take the mean of the means, because this would result in the VisA dataset classes having

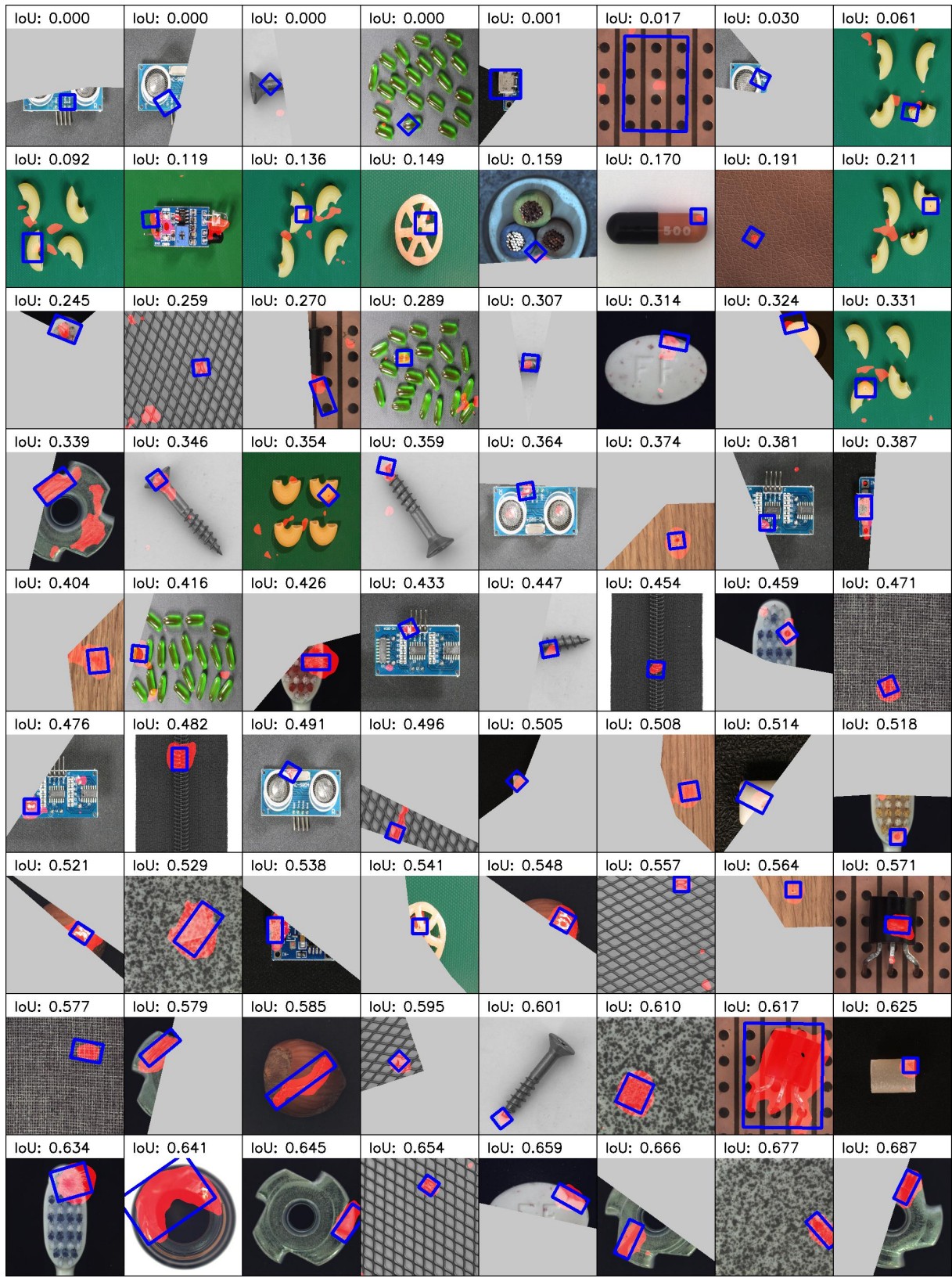

Figure 5: Demonstration of ordered IoU values between prediction (red) and bounding box label (blue). Random images from various MVTec and VisA classes, with predictions created using the PatchCore algorithm (Roth et al., 2022). We recommend an IoU limit of 0.3, as we propose that from this point onwards (the third row), the prediction and bounding box have satisfactory overlap. Note the gray area means ignored pixels, due to there being other closer anomalies in the given image (see above for explanation).

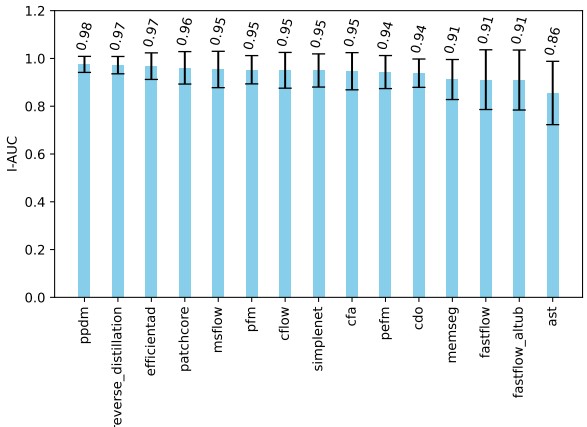

(a) Algorithm rankings based on the image-wise AUC metric.

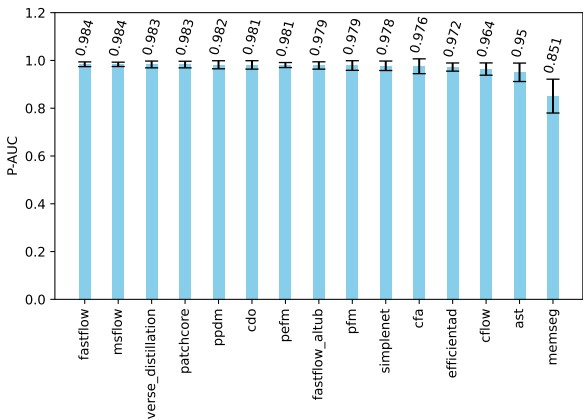

(b) Algorithm rankings based on the pixel-wise AUC metric.

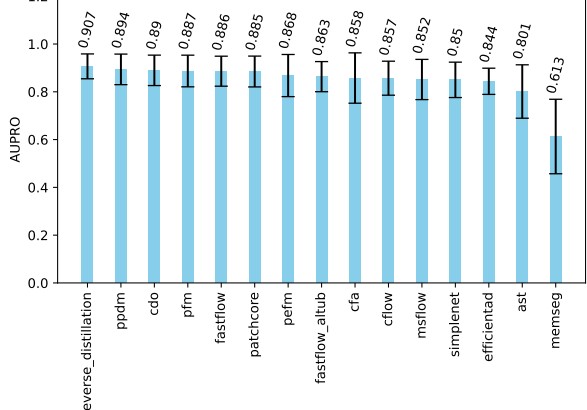

(c) Algorithm rankings based on the AUPRO metric.

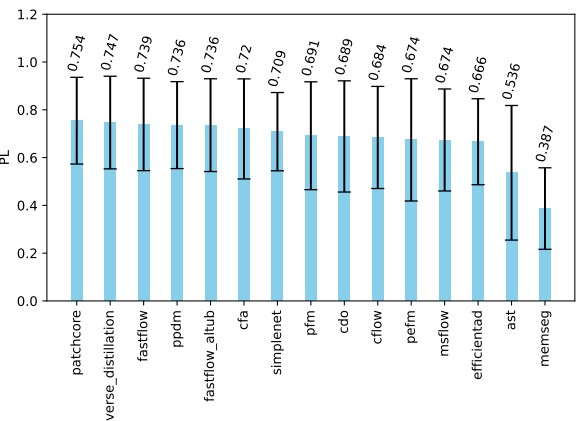

(d) Algorithm rankings based on the PL metric.

Figure 6: We present the VADI results. This is the mean of the pooled classes of the MVTec and VisA datasets (each class of each dataset weighted equally). As a measure of variance, we also show the standard deviation of the result across each class as error bars.

a greater weighting due to the lower number of classes in VisA (12) compared to MVTec (15). We propose that the VADI benchmark produces the most comprehensive and fair set of results currently available for industrial anomaly detection.

We trialled the top 15 anomaly detection algorithms which were selected by the criteria described in Section 2.1. Each algorithm was given a training time of two hours (evaluation and metric calculation is not counted). There were no extra image augmentations carried out in training (we leave a similar run with maximal training augmentations for future work). Each algorithm was run using the default parameters from the publication. The hyperparameters were kept the same across each class (this may disadvantage some algorithms which change hyperparameters based on class category, e.g. texture or object). We do this because we believe that amending hyperparameters for a certain class makes use of forbidden test set information. When an industry professional is presented with a new problem, they would not have the luxury of amending the parameters based on the test set results (unless a large number of labelled anomalies are available).

Figure 6 shows the VADI scores for each algorithm and metric. Tables presenting the class breakdown of these results across all algorithms are shown in Appendix B.

The VADI benchmark finds that PPDM, Fastflow, Reverse Distillation, and PatchCore are the best anomaly detection algorithms for I-AUC, P-AUC, AUPRO, and PL respectively. Due to the advantages of PL mentioned above, we use PL version as final ranking. Therefore we find PatchCore to be the best algorithm currently available. We note that according to the PWC publicly available leaderboard (MetaAI, 2022), PatchCore is in the 11[th] position for I-AUC, and 23[rd] position for P-AUC, with no result for AUPRO (April 2024). We use this example to show the importance of not relying on self-reported results to judge algorithms.

The error bars of Figure 6 show the standard deviation taken across the classes. The standard deviation bars of P-AUC show a small output of this metric as the bars are very small. The error bars show that PL has the greatest range of output, highlighting an advantage of PL.

A final takeaway from Figure 6 is that the performance of the algorithms are relatively close to each other. There is not one dominant algorithm in the field of anomaly detection. This implies that all these families of algorithms warrant further research.

## 5 Conclusion and Future Work

We release the VisionAD anomaly detection library intended for fair benchmarking and algorithm development. We also release a novel metric Proportion Localised, intended to give researchers a more intuitive metric by classifying each anomaly as localised or not. We release the VisionADIndustrial (VADI) benchmark, ranking the top 15 algorithms available on the traditional and introduced metric(s) via the performance on the MVTec and VisA datasets. We find that PatchCore scores the highest for the PL metric, and therefore we find that PatchCore is the current best algorithm for industrial anomaly detection.

A limitation of VisionAD is that many algorithms are formatted to work in one system. This can mean certain algorithms are written less naturally than they may be written when standing alone. Formatting different algorithms to fit into one API pattern is a necessary compromise to achieve the benefits VisionAD offers. A limitation of the VADI benchmark is the lack of hyperparameter sweeping and training augmentations (although we used the recommended hyperparameters for each algorithm). We leave longer experiments, training augmentations, and hyperparameter tuning to future iterations of the VisionAD leaderboard.

Future work includes adding more algorithms to VisionAD. Features could be added such as memory logging and a results visualisation dashboard. VisionAD is backed by a research group committed to improving and maintaining the library.

Funded by Rolls-Royce Defence; supported by UK Research and Innovation (UKRI), grant reference number EP/S023437/1.

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

## Appendix A

### Adding algorithms

The ease of adding algorithms is intended to be one of the major strengths of VisionAD. The process is described below.

Each algorithm is built from a class called {*algo_name*}Wrapper, which inherits the ModelWrapper class. Using the provided template, a researcher only needs to implement six methods for the algorithm to work. Each method is discussed below:

**Initialisation: __init__(self)**
Initialise the model and optimisers.

**Training: train_one_epoch(self)**
Train the model using the self.dataloader_train (automatically attached via the wrapper).

**Pre-eval processing: prev_eval_processing(self)**
Optionally undertake any necessary pre-processing before evaluation, using access to self.dataloader_train. Left as 'pass' in most algorithms. This cannot be done at the start of eval_outputs_dataloader, as eval_outputs_dataloader may be called on a new class instance in the case of loading the model from disk, where the train generator would not be available.

**Evaluation: eval_outputs_dataloader(self, generator, len_generator)**
This method is called twice, once with a generator of regular test images, and once with a generator of anomalous test images. To avoid data leakage, this method does not have access to ground truths or paths. The method must iterate over the passed generator, saving to memory an anomaly map and score for each image. Returns a tuple of two dictionaries: a dictionary of anomaly maps: {*'anomaly_map_method_a'*: *torch.tensor/np.array of shape (no.images, height, width)*}, and a dictionary image scores *torch.tensor/np.array of shape (no.images)*}. Notably, the system allows any number of different methods for creating image scores and heatmaps for a given algorithm. For instance, the researcher may wish to trial reducing a feature channel via mean, standard deviation, and max, without wishing to rewrite three algorithms. The metric code will provide results for each key and tensor passed through.

**Saving: save_model(self, location)**
Saves the parameters of the model to a given location.

**Loading: load_model(self, location)**
Load the parameters of the model from a given location.

A trivial example of a random classifier is included to help researchers with the small but necessary template code.

### Metrics

VisionAD has a comprehensive list of metrics, including the standard I-AUC, P-AUC, and AUPRO, alongside the introduced PL. However, researchers can add metrics as they desire. A researcher only needs to implement a function which accepts an array of targets, predictions, labels and image scores, and returns a single value.

### Run configuration files

Config files are Python files. Each file contains a model_list variable, which is a list of dictionaries each containing the information to run a model (algo_name, model_parameters, epoch, n_epochs_test). Each file also contains a dataset_list, which contains strings which match the dataset keys (discussed below). The wrapper will run each model on each dataset in a given config file.

The choice of Python config file over Yaml is done because it gives the researcher the expressiveness of Python when creating experiments. For instance, each algorithm contains a default set of model_parameters, and these can be imported into different config files whenever this algorithm is run (with changes made when

necessary). Looping can also allow one parameter to be changed whilst other are kept the same via copying the model dictionary inside model_list X times, but changing the target hyperparameter and description each time. Many other similar tricks are available which save the researcher the painful process of copy and pasting parameters.

Basic and complex examples of configuration files are given to allow researchers to get started quickly.

### Data configuration files

The dataset file datasets/configure_dataset contains a variable called datasets, which is a dictionary of dataset keys, where the item of each key is a dictionary with various dataset information (training image path, testing image path, ...). These should be self explanatory. These dataset keys should match the strings in the run configuration files. To add a dataset a user needs to add a dataset key and fill the necessary paths.

### Command line use

The wrapper is designed to be as easy as possible to run, with minimal command line arguments. The call *python3 run.py –config configs/my_experiment.py –device 'cuda:0'* is sufficient to load a config file and overwrite the gpu device. The config file contains everything else needed.

### IPython experimentation

Enabling easy experimentation is a goal of VisionAD. Early stage algorithm development is often done in an IPython environment. The necessary template code can be loaded into a IPython environment such as Jupyter notebook. The researcher can experiment with the algorithm class whilst the data loading and evaluation code is handled. A starter notebook is provided to demonstrate this using the trivial random classifier.

### Other features

As mentioned above, the library does not give algorithms access to the ground truths during training, to ensure data leakage is not possible. The wrapper also contains a number of other bug checking features such as ensuring the outputs of the algorithms are the right dimensions. The wrapper also measures total training time, training time per image, total inference time, and inference time per image. The wrapper allows all metrics to be logged to Weights and Biases (Wandb) if desired, and code is also provided to pull these results from Wandb and parse them into the tables shown in this publication.

Some algorithms require synthetic anomalies. To facilitate this, we allow a callback to be ran on the training data before it is sent through the dataloader. The training dataloader outputs three items, the training image, the result of this callback, and the image file name. The default of the callback result is to return 0, which would be passed to the algorithm and ignored. However in the case that synthetic anomalies are added to the training images, this callback could return the corresponding mask, which the algorithm can use.

All VisionAD algorithms work on Windows and Linux, CPU and GPU. Currently all algorithms fit on an RTX 3090 24Gb GPU.

## Appendix B

### Image-wise AUC - MVTec

Table 2 we show the results using the image-wise AUC metric on the MVTec dataset.

|  | efficientad | reverse distillation | fastflow | msflow | ppdm | fastflow altub | patchcore |
|---|---|---|---|---|---|---|---|
| bottle | 0.999 | **1** | **1** | **1** | **1** | **1** | **1** |
| cable | 0.979 | 0.982 | 0.986 | 0.997 | 0.979 | 0.98 | **0.998** |
| capsule | 0.987 | 0.984 | **0.995** | 0.991 | 0.982 | 0.99 | 0.991 |
| carpet | 0.987 | **1** | 0.988 | 0.987 | 0.997 | 0.989 | 0.978 |
| grid | 0.998 | 0.994 | **1** | 0.99 | 0.927 | **1** | 0.994 |
| hazelnut | **1** | **1** | **1** | **1** | **1** | **1** | **1** |
| leather | **1** | **1** | **1** | **1** | **1** | **1** | **1** |
| metal nut | **1** | **1** | 0.999 | 0.997 | **1** | **1** | **1** |
| pill | **0.996** | 0.974 | 0.979 | 0.962 | 0.968 | 0.965 | 0.975 |
| screw | 0.945 | 0.986 | 0.945 | 0.926 | 0.982 | 0.923 | 0.974 |
| tile | **1** | **1** | **1** | **1** | **1** | **1** | **1** |
| toothbrush | **1** | **1** | 0.981 | **1** | **1** | 0.975 | 0.914 |
| transistor | **1** | 0.977 | 0.998 | 0.999 | **1** | **1** | 0.995 |
| wood | 0.998 | 0.996 | 0.998 | **1** | **1** | **1** | 0.993 |
| zipper | **0.997** | 0.982 | 0.996 | **0.997** | 0.992 | 0.993 | 0.995 |
| mean | **0.992** | **0.992** | 0.991 | 0.99 | 0.988 | 0.988 | 0.987 |

|  | cfa | pfm | pefm | cflow | simplenet | memseg | cdo | ast |
|---|---|---|---|---|---|---|---|---|
| bottle | 0.998 | **1** | **1** | 0.995 | **1** | **1** | **1** | 0.987 |
| cable | **0.998** | 0.996 | 0.996 | 0.978 | 0.993 | 0.957 | 0.908 | 0.957 |
| capsule | 0.958 | 0.958 | 0.945 | 0.963 | 0.99 | 0.97 | 0.83 | 0.895 |
| carpet | 0.984 | 0.99 | 0.996 | 0.999 | 0.976 | 0.927 | 0.999 | 0.994 |
| grid | 0.96 | 0.991 | 0.992 | 0.953 | 0.728 | 0.938 | 0.952 | 0.759 |
| hazelnut | 0.998 | **1** | **1** | 0.998 | **1** | 0.992 | 0.991 | 0.959 |
| leather | **1** | **1** | **1** | **1** | **1** | **1** | **1** | **1** |
| metal nut | 0.999 | **1** | **1** | 0.997 | 0.999 | **1** | 0.992 | 0.965 |
| pill | 0.986 | 0.976 | 0.987 | 0.948 | 0.974 | 0.91 | 0.918 | 0.877 |
| screw | 0.868 | 0.962 | 0.915 | 0.934 | 0.885 | 0.997 | **0.999** | 0.735 |
| tile | **1** | 0.997 | 0.997 | 0.999 | **1** | **1** | **1** | 0.996 |
| toothbrush | **1** | 0.886 | 0.892 | 0.967 | 0.906 | **1** | 0.853 | 0.961 |
| transistor | 0.999 | 0.978 | 0.996 | 0.904 | **1** | 0.969 | 0.921 | 0.976 |
| wood | 0.988 | 0.996 | 0.997 | 0.992 | 0.995 | 0.999 | 0.992 | 0.971 |
| zipper | 0.985 | 0.969 | 0.97 | 0.975 | 0.996 | 0.778 | 0.971 | 0.98 |
| mean | 0.981 | 0.98 | 0.979 | 0.973 | 0.963 | 0.962 | 0.955 | 0.934 |

Table 2: VisionAD results on the MVTec dataset based on the image-wise AUC metric.

**Pixel-wise AUC - MVTec**

Table 3 we show the results using the pixel-wise AUC metric on the MVTec dataset.

| | cfa | fastflow | msflow | patchcore | pefm | reverse distillation | cflow |
|---|---|---|---|---|---|---|---|
| bottle | 0.99 | 0.989 | 0.985 | 0.988 | 0.985 | 0.987 | 0.987 |
| cable | **0.989** | 0.977 | 0.986 | 0.986 | 0.98 | 0.979 | 0.975 |
| capsule | 0.99 | **0.991** | 0.989 | **0.991** | 0.983 | 0.987 | 0.99 |
| carpet | 0.985 | 0.989 | 0.99 | 0.99 | 0.99 | **0.992** | 0.991 |
| grid | 0.978 | **0.993** | 0.981 | 0.986 | 0.988 | 0.991 | 0.974 |
| hazelnut | 0.987 | 0.983 | 0.982 | 0.989 | 0.991 | 0.991 | 0.989 |
| leather | 0.995 | **0.996** | **0.996** | 0.995 | 0.994 | 0.995 | **0.996** |
| metal nut | 0.985 | 0.988 | 0.987 | **0.99** | 0.974 | 0.976 | 0.985 |
| pill | 0.985 | 0.978 | 0.988 | 0.981 | 0.979 | 0.984 | **0.989** |
| screw | 0.986 | 0.989 | 0.983 | 0.994 | 0.991 | **0.997** | 0.988 |
| tile | **0.982** | 0.968 | 0.981 | 0.972 | 0.962 | 0.964 | 0.967 |
| toothbrush | **0.992** | 0.983 | 0.982 | 0.989 | 0.988 | 0.991 | 0.984 |
| transistor | 0.984 | 0.98 | 0.983 | 0.937 | 0.973 | 0.936 | 0.948 |
| wood | 0.962 | 0.969 | 0.952 | 0.959 | 0.96 | 0.959 | 0.953 |
| zipper | **0.99** | 0.986 | 0.985 | **0.99** | 0.984 | 0.987 | 0.979 |
| mean | **0.985** | 0.984 | 0.983 | 0.982 | 0.981 | 0.981 | 0.98 |

| | cdo | ppdm | fastflow altub | efficientad | simplenet | pfm | ast | memseg |
|---|---|---|---|---|---|---|---|---|
| bottle | **0.991** | 0.989 | 0.98 | 0.957 | 0.981 | 0.984 | 0.928 | 0.958 |
| cable | 0.97 | 0.981 | 0.964 | 0.984 | 0.975 | 0.969 | 0.961 | 0.794 |
| capsule | 0.98 | 0.982 | 0.987 | 0.985 | 0.99 | 0.985 | 0.97 | 0.926 |
| carpet | **0.992** | **0.992** | 0.985 | 0.964 | 0.984 | 0.989 | 0.984 | 0.93 |
| grid | 0.986 | 0.916 | 0.99 | 0.976 | 0.902 | 0.988 | 0.97 | 0.895 |
| hazelnut | **0.992** | 0.991 | 0.974 | 0.978 | 0.982 | 0.991 | 0.981 | 0.826 |
| leather | **0.996** | 0.993 | 0.995 | 0.991 | 0.994 | 0.994 | 0.985 | 0.988 |
| metal nut | 0.985 | 0.98 | 0.978 | 0.967 | 0.988 | 0.975 | 0.959 | 0.771 |
| pill | 0.983 | 0.983 | 0.974 | 0.982 | 0.986 | 0.974 | 0.953 | 0.887 |
| screw | 0.985 | 0.995 | 0.979 | 0.987 | 0.99 | 0.99 | 0.983 | 0.885 |
| tile | 0.974 | 0.962 | 0.95 | 0.979 | 0.965 | 0.963 | 0.937 | 0.968 |
| toothbrush | 0.988 | **0.992** | 0.98 | 0.98 | 0.985 | 0.987 | 0.984 | 0.965 |
| transistor | 0.911 | 0.965 | 0.974 | **0.985** | 0.976 | 0.891 | 0.927 | 0.706 |
| wood | **0.973** | 0.956 | 0.948 | 0.953 | 0.942 | 0.961 | 0.923 | 0.909 |
| zipper | 0.982 | 0.977 | 0.979 | 0.961 | 0.984 | 0.983 | 0.967 | 0.891 |
| mean | 0.979 | 0.977 | 0.976 | 0.975 | 0.975 | 0.975 | 0.961 | 0.887 |

Table 3: VisionAD results on the MVTec dataset based on the pixel-wise AUC metric.

**AUPRO - MVTec**

Table 4 we show the results using the AUPRO metric on the MVTec dataset.

|  | reverse distillation | pefm | cfa | cdo | fastflow | patchcore | pfm |
|---|---|---|---|---|---|---|---|
| bottle | 0.949 | 0.941 | 0.941 | 0.951 | 0.924 | 0.938 | 0.934 |
| cable | 0.913 | 0.91 | 0.909 | 0.89 | 0.84 | **0.914** | 0.893 |
| capsule | 0.923 | 0.909 | 0.936 | 0.909 | **0.938** | 0.937 | 0.922 |
| carpet | 0.957 | 0.957 | 0.939 | **0.962** | 0.946 | 0.943 | 0.95 |
| grid | 0.955 | 0.953 | 0.897 | 0.95 | **0.956** | 0.92 | 0.951 |
| hazelnut | 0.95 | **0.955** | 0.926 | 0.949 | 0.941 | 0.943 | 0.952 |
| leather | 0.979 | 0.983 | 0.972 | 0.984 | **0.988** | 0.962 | 0.985 |
| metal nut | 0.941 | 0.937 | 0.923 | **0.942** | 0.912 | 0.94 | 0.933 |
| pill | **0.967** | 0.961 | 0.961 | 0.964 | 0.94 | 0.956 | 0.96 |
| screw | **0.978** | 0.94 | 0.922 | 0.92 | 0.932 | 0.966 | 0.942 |
| tile | 0.846 | 0.841 | **0.901** | 0.893 | 0.875 | 0.861 | 0.845 |
| toothbrush | 0.912 | 0.878 | 0.905 | 0.873 | 0.879 | 0.875 | 0.878 |
| transistor | 0.805 | 0.843 | 0.884 | 0.768 | 0.869 | 0.808 | 0.714 |
| wood | 0.909 | 0.924 | 0.892 | 0.932 | **0.933** | 0.883 | 0.906 |
| zipper | 0.946 | 0.937 | 0.951 | 0.931 | 0.933 | **0.954** | 0.937 |
| mean | **0.929** | 0.925 | 0.924 | 0.921 | 0.92 | 0.92 | 0.913 |

|  | ppdm | cflow | msflow | fastflow altub | simplenet | efficientad | ast | memseg |
|---|---|---|---|---|---|---|---|---|
| bottle | **0.952** | 0.903 | 0.899 | 0.881 | 0.871 | 0.782 | 0.814 | 0.786 |
| cable | 0.903 | 0.877 | 0.866 | 0.813 | 0.872 | 0.841 | 0.822 | 0.425 |
| capsule | 0.889 | 0.918 | 0.906 | 0.923 | 0.916 | 0.916 | 0.841 | 0.594 |
| carpet | 0.952 | 0.95 | 0.932 | 0.936 | 0.901 | 0.836 | 0.935 | 0.704 |
| grid | 0.7 | 0.895 | 0.897 | 0.937 | 0.709 | 0.901 | 0.848 | 0.43 |
| hazelnut | 0.945 | 0.94 | 0.917 | 0.931 | 0.91 | 0.905 | 0.876 | 0.652 |
| leather | 0.972 | 0.984 | 0.977 | 0.985 | 0.955 | 0.94 | 0.954 | 0.786 |
| metal nut | **0.942** | 0.883 | 0.863 | 0.89 | 0.894 | 0.778 | 0.857 | 0.569 |
| pill | 0.953 | 0.946 | 0.935 | 0.907 | 0.933 | 0.889 | 0.802 | 0.197 |
| screw | 0.968 | 0.938 | 0.894 | 0.84 | 0.941 | 0.934 | 0.902 | 0.197 |
| tile | 0.829 | 0.837 | 0.88 | 0.831 | 0.822 | 0.879 | 0.79 | 0.734 |
| toothbrush | **0.913** | 0.856 | 0.812 | 0.795 | 0.826 | 0.831 | 0.824 | 0.524 |
| transistor | 0.822 | 0.775 | 0.864 | 0.784 | 0.824 | **0.899** | 0.748 | 0.401 |
| wood | 0.873 | 0.898 | 0.872 | 0.917 | 0.814 | 0.803 | 0.847 | 0.686 |
| zipper | 0.925 | 0.889 | 0.928 | 0.901 | 0.922 | 0.813 | 0.865 | 0.574 |
| mean | 0.903 | 0.899 | 0.896 | 0.885 | 0.874 | 0.863 | 0.848 | 0.551 |

Table 4: VisionAD results on the MVTec dataset based on the AUPRO metric.

**PL** - **MVTec**

Table 5 we show the results using the introduced PL metric on the MVTec dataset.

|  | reverse distillation | patchcore | fastflow | fastflow altub | cfa | pefm | cdo |
|---|---|---|---|---|---|---|---|
| bottle | 0.971 | 0.971 | 0.956 | 0.965 | 0.956 | 0.971 | **0.985** |
| cable | 0.904 | **0.936** | 0.8 | 0.805 | 0.912 | 0.928 | 0.832 |
| capsule | 0.763 | **0.842** | 0.798 | 0.806 | 0.754 | 0.57 | 0.544 |
| carpet | 0.914 | 0.903 | 0.914 | 0.928 | 0.882 | 0.925 | **0.946** |
| grid | 0.897 | 0.868 | **0.934** | 0.931 | 0.846 | 0.669 | 0.86 |
| hazelnut | **0.958** | 0.938 | 0.917 | 0.901 | 0.906 | 0.948 | 0.948 |
| leather | 0.979 | 0.895 | 1 | 1.02 | 0.905 | 0.989 | 1 |
| metal nut | **0.992** | 0.975 | 0.967 | 0.938 | 0.942 | 0.975 | 0.967 |
| pill | 0.754 | 0.856 | 0.814 | 0.79 | **0.868** | **0.868** | 0.832 |
| screw | 0.722 | **0.746** | 0.635 | 0.634 | 0.516 | 0.452 | 0.5 |
| tile | 0.966 | 0.966 | 0.966 | 0.946 | **0.977** | 0.966 | 0.966 |
| toothbrush | 0.846 | 0.827 | 0.788 | 0.798 | 0.865 | 0.808 | 0.788 |
| transistor | 0.837 | 0.744 | 0.93 | **0.932** | 0.907 | 0.884 | 0.651 |
| wood | 0.916 | 0.877 | 0.961 | **0.971** | 0.91 | 0.923 | 0.942 |
| zipper | **0.926** | **0.926** | 0.869 | 0.854 | 0.898 | 0.858 | 0.875 |
| mean | **0.89** | 0.885 | 0.883 | 0.881 | 0.87 | 0.849 | 0.842 |

|  | pfm | cflow | msflow | ppdm | simplenet | efficientad | ast | memseg |
|---|---|---|---|---|---|---|---|---|
| bottle | 0.971 | 0.926 | 0.941 | **0.985** | 0.926 | 0.882 | 0.662 | 0.787 |
| cable | 0.864 | 0.76 | 0.88 | 0.912 | 0.88 | 0.712 | 0.758 | 0.378 |
| capsule | 0.702 | 0.544 | 0.667 | 0.623 | 0.807 | 0.798 | 0.395 | 0.235 |
| carpet | 0.892 | 0.914 | 0.914 | 0.935 | 0.849 | 0.882 | 0.903 | 0.22 |
| grid | 0.691 | 0.64 | 0.787 | 0.272 | 0.485 | 0.904 | 0.679 | 0.168 |
| hazelnut | **0.958** | 0.914 | 0.812 | 0.938 | 0.875 | 0.896 | 0.884 | 0.353 |
| leather | 0.989 | **1.05** | 0.968 | 0.947 | 0.811 | 0.884 | 0.937 | 0.51 |
| metal nut | 0.983 | 0.946 | 0.942 | 0.975 | 0.942 | 0.867 | 0.899 | 0.381 |
| pill | 0.85 | 0.763 | 0.731 | 0.844 | 0.772 | 0.653 | 0.461 | 0.159 |
| screw | 0.389 | 0.587 | 0.452 | 0.603 | 0.54 | 0.484 | 0.37 | 0.00719 |
| tile | 0.966 | 0.941 | 0.966 | 0.966 | 0.943 | 0.898 | 0.942 | 0.598 |
| toothbrush | 0.808 | **0.899** | 0.692 | 0.865 | 0.788 | 0.769 | 0.692 | 0.42 |
| transistor | 0.698 | 0.777 | 0.93 | 0.744 | 0.86 | 0.535 | 0.721 | 0.295 |
| wood | 0.877 | 0.92 | 0.91 | 0.903 | 0.819 | 0.774 | 0.865 | 0.236 |
| zipper | 0.903 | 0.853 | 0.835 | 0.778 | 0.812 | 0.608 | 0.67 | 0.167 |
| mean | 0.836 | 0.829 | 0.828 | 0.819 | 0.807 | 0.77 | 0.723 | 0.328 |

Table 5: VisionAD results on the MVTec dataset based on the introduced PL metric.

**Image-wise AUC** - **VisA**

Table 6 we show the results using the image-wise AUC metric on the VisA dataset.

|  | ppdm | reverse distillation | efficientad | simplenet | patchcore | cflow | pfm |
|---|---|---|---|---|---|---|---|
| pipe fryum | 0.994 | 0.993 | 0.721 | 0.99 | **0.996** | 0.95 | **0.996** |
| pcb4 | 0.996 | **0.999** | 0.979 | 0.981 | 0.994 | 0.975 | 0.977 |
| pcb3 | 0.949 | 0.952 | 0.923 | 0.936 | **0.968** | 0.712 | 0.959 |
| pcb2 | **0.982** | 0.94 | 0.956 | 0.959 | 0.956 | 0.82 | 0.929 |
| pcb1 | 0.977 | 0.969 | 0.977 | 0.979 | 0.98 | **1.01** | 0.921 |
| macaroni2 | **0.917** | 0.85 | 0.897 | 0.759 | 0.719 | 0.805 | 0.799 |
| macaroni1 | **0.982** | 0.968 | 0.972 | 0.935 | 0.93 | 0.929 | 0.907 |
| fryum | 0.947 | 0.914 | 0.978 | 0.934 | 0.932 | **0.988** | 0.972 |
| chewinggum | 0.983 | 0.99 | 0.988 | 0.978 | 0.979 | **1.06** | 0.99 |
| cashew | 0.966 | 0.964 | 0.964 | 0.912 | 0.968 | 0.967 | 0.903 |
| capsules | 0.849 | 0.903 | **0.917** | 0.863 | 0.751 | 0.831 | 0.766 |
| candle | 0.959 | 0.925 | 0.971 | 0.973 | 0.958 | **1.01** | 0.905 |
| mean | **0.958** | 0.947 | 0.937 | 0.933 | 0.928 | 0.921 | 0.919 |

|  | cdo | msflow | cfa | pefm | memseg | fastflow altub | fastflow | ast |
|---|---|---|---|---|---|---|---|---|
| pipe fryum | 0.96 | 0.972 | 0.976 | 0.992 | 0.837 | 0.678 | 0.683 | 0.689 |
| pcb4 | 0.953 | 0.988 | 0.99 | 0.979 | 0.934 | 0.977 | 0.968 | 0.96 |
| pcb3 | 0.849 | 0.909 | 0.91 | 0.923 | 0.839 | 0.805 | 0.825 | 0.774 |
| pcb2 | 0.95 | 0.955 | 0.872 | 0.902 | 0.874 | 0.88 | 0.865 | 0.89 |
| pcb1 | 0.938 | 0.959 | 0.994 | 0.907 | 0.887 | 0.884 | 0.885 | 0.735 |
| macaroni2 | 0.807 | 0.683 | 0.719 | 0.727 | 0.709 | 0.629 | 0.621 | 0.584 |
| macaroni1 | 0.93 | 0.884 | 0.864 | 0.875 | 0.819 | 0.724 | 0.725 | 0.581 |
| fryum | 0.936 | 0.949 | 0.918 | 0.925 | 0.871 | 0.936 | 0.92 | 0.614 |
| chewinggum | 0.975 | 0.986 | 0.998 | 0.971 | 0.934 | 1 | 0.985 | 0.854 |
| cashew | **0.975** | 0.904 | 0.919 | 0.896 | 0.852 | 0.773 | 0.776 | 0.745 |
| capsules | 0.809 | 0.74 | 0.715 | 0.767 | 0.732 | 0.569 | 0.571 | 0.795 |
| candle | 0.926 | 0.975 | 0.958 | 0.914 | 0.893 | 0.891 | 0.913 | 0.864 |
| mean | 0.917 | 0.909 | 0.903 | 0.898 | 0.848 | 0.812 | 0.811 | 0.757 |

Table 6: VisionAD results on the Visa dataset based on the image-wise AUC metric.

**Pixel-wise AUC** - **VisA**

Table 7 we show the results using the pixel-wise AUC metric on the VisA dataset.

|  | ppdm | reverse distillation | fastflow | msflow | cdo | pfm | patchcore |
|---|---|---|---|---|---|---|---|
| pipe fryum | 0.992 | 0.99 | 0.99 | **0.993** | 0.99 | 0.99 | 0.992 |
| pcb4 | **0.988** | 0.985 | 0.983 | 0.98 | 0.986 | 0.968 | 0.977 |
| pcb3 | 0.992 | 0.992 | 0.989 | 0.978 | 0.987 | 0.99 | 0.99 |
| pcb2 | **0.99** | 0.987 | 0.984 | 0.981 | 0.981 | 0.987 | 0.988 |
| pcb1 | **0.998** | 0.997 | 0.995 | 0.996 | 0.995 | 0.996 | 0.997 |
| macaroni2 | **0.996** | 0.994 | 0.984 | 0.98 | 0.992 | 0.991 | 0.969 |
| macaroni1 | **0.997** | 0.996 | 0.994 | 0.99 | 0.991 | 0.992 | 0.991 |
| fryum | **0.97** | 0.964 | 0.947 | 0.961 | 0.938 | 0.957 | 0.952 |
| chewinggum | 0.985 | 0.984 | 0.988 | 0.983 | 0.987 | 0.993 | 0.986 |
| cashew | 0.978 | 0.961 | 0.984 | **0.992** | 0.982 | 0.981 | 0.987 |
| capsules | **0.994** | 0.993 | 0.987 | 0.987 | 0.987 | 0.975 | 0.978 |
| candle | 0.975 | 0.987 | 0.985 | 0.987 | 0.988 | 0.98 | **0.992** |
| mean | **0.988** | 0.986 | 0.984 | 0.984 | 0.984 | 0.983 | 0.983 |

|  | simplenet | fastflow altub | pefm | efficientad | cfa | cflow | ast | memseg |
|---|---|---|---|---|---|---|---|---|
| pipe fryum | 0.991 | 0.988 | 0.992 | 0.98 | **0.993** | 0.951 | 0.984 | 0.834 |
| pcb4 | 0.977 | 0.972 | 0.978 | 0.987 | 0.909 | 0.983 | 0.985 | 0.824 |
| pcb3 | 0.984 | **0.996** | 0.989 | 0.986 | 0.987 | 0.979 | 0.962 | 0.833 |
| pcb2 | 0.982 | 0.972 | 0.986 | 0.965 | 0.984 | 0.975 | 0.981 | 0.829 |
| pcb1 | 0.997 | 0.996 | 0.995 | 0.993 | 0.992 | 0.947 | 0.876 | 0.831 |
| macaroni2 | 0.96 | 0.96 | 0.98 | 0.959 | 0.938 | 0.932 | 0.906 | 0.817 |
| macaroni1 | 0.992 | 0.994 | 0.988 | 0.977 | 0.994 | 0.946 | 0.903 | 0.829 |
| fryum | 0.944 | 0.948 | 0.963 | 0.91 | 0.848 | 0.902 | 0.903 | 0.791 |
| chewinggum | 0.986 | **1.01** | 0.98 | 0.985 | 0.988 | 0.945 | 0.941 | 0.829 |
| cashew | 0.988 | 0.98 | 0.961 | 0.954 | 0.988 | 0.929 | 0.835 | 0.814 |
| capsules | 0.986 | 0.979 | 0.964 | 0.971 | 0.977 | 0.943 | 0.988 | 0.827 |
| candle | 0.985 | 0.98 | 0.98 | 0.956 | 0.966 | 0.942 | 0.984 | 0.826 |
| mean | 0.981 | 0.981 | 0.98 | 0.969 | 0.964 | 0.948 | 0.937 | 0.824 |

Table 7: VisionAD results on the VisA dataset based on the pixel-wise AUC metric.

**AUPRO** - **VisA**

Table 8 we show the results using the AUPRO metric on the VisA dataset.

| | ppdm | reverse distillation | pfm | cdo | fastflow | patchcore | fastflow altub |
|---|---|---|---|---|---|---|---|
| pipe fryum | **0.931** | 0.927 | 0.906 | 0.9 | 0.855 | 0.921 | 0.843 |
| pcb4 | 0.847 | 0.811 | 0.713 | 0.83 | 0.758 | 0.734 | 0.769 |
| pcb3 | 0.86 | **0.861** | 0.852 | 0.8 | 0.808 | 0.817 | 0.808 |
| pcb2 | **0.864** | 0.846 | 0.823 | 0.791 | 0.795 | 0.852 | 0.774 |
| pcb1 | **0.931** | 0.899 | 0.883 | 0.88 | 0.849 | 0.873 | 0.849 |
| macaroni2 | **0.956** | 0.93 | 0.895 | 0.924 | 0.884 | 0.823 | 0.867 |
| macaroni1 | **0.942** | 0.921 | 0.901 | 0.884 | 0.927 | 0.899 | 0.909 |
| fryum | 0.823 | **0.833** | **0.833** | 0.788 | 0.715 | 0.787 | 0.721 |
| chewinggum | 0.767 | 0.798 | 0.842 | 0.749 | 0.847 | 0.789 | **0.85** |
| cashew | **0.912** | 0.898 | 0.894 | 0.891 | 0.866 | 0.888 | 0.862 |
| capsules | 0.877 | **0.912** | 0.806 | 0.846 | 0.899 | 0.78 | 0.904 |
| candle | 0.887 | 0.911 | 0.903 | 0.925 | 0.923 | **0.931** | 0.929 |
| mean | **0.883** | 0.879 | 0.854 | 0.851 | 0.844 | 0.841 | 0.84 |

| | simplenet | efficientad | cflow | pefm | msflow | cfa | ast | memseg |
|---|---|---|---|---|---|---|---|---|
| pipe fryum | 0.845 | 0.882 | 0.863 | 0.897 | 0.85 | 0.875 | 0.898 | 0.762 |
| pcb4 | 0.734 | **0.86** | 0.796 | 0.754 | 0.733 | 0.65 | 0.841 | 0.666 |
| pcb3 | 0.701 | 0.784 | 0.714 | 0.826 | 0.646 | 0.828 | 0.517 | 0.665 |
| pcb2 | 0.716 | 0.805 | 0.734 | 0.816 | 0.736 | 0.806 | 0.785 | 0.685 |
| pcb1 | 0.887 | 0.848 | 0.808 | 0.853 | 0.824 | 0.774 | 0.443 | 0.714 |
| macaroni2 | 0.892 | 0.804 | 0.839 | 0.794 | 0.929 | 0.771 | 0.736 | 0.741 |
| macaroni1 | 0.934 | 0.869 | 0.87 | 0.865 | 0.913 | 0.931 | 0.712 | 0.768 |
| fryum | 0.747 | 0.726 | 0.721 | 0.764 | 0.645 | 0.525 | 0.739 | 0.637 |
| chewinggum | 0.812 | 0.764 | 0.765 | 0.651 | 0.788 | 0.832 | 0.708 | 0.675 |
| cashew | 0.845 | 0.877 | 0.821 | 0.764 | 0.781 | 0.738 | 0.752 | 0.725 |
| capsules | 0.82 | 0.782 | 0.798 | 0.664 | 0.791 | 0.682 | 0.884 | 0.705 |
| candle | 0.908 | 0.84 | 0.882 | 0.921 | 0.914 | 0.887 | 0.893 | 0.779 |
| mean | 0.82 | 0.82 | 0.801 | 0.797 | 0.796 | 0.775 | 0.742 | 0.71 |

Table 8: VisionAD results on the VisA dataset based on the AUPRO metric.

**PL** - **VisA**

Table 9 we show the results using the introduced PL metric on the VisA dataset.

|  | ppdm | patchcore | simplenet | reverse distillation | fastflow | fastflow altub | efficientad |
|---|---|---|---|---|---|---|---|
| pipe fryum | 0.806 | 0.823 | 0.774 | 0.798 | 0.774 | 0.759 | 0.774 |
| pcb4 | 0.696 | 0.675 | 0.613 | 0.639 | 0.629 | 0.62 | 0.706 |
| pcb3 | 0.592 | **0.6** | 0.424 | 0.584 | 0.504 | 0.505 | 0.568 |
| pcb2 | 0.606 | **0.642** | 0.489 | 0.496 | 0.482 | 0.474 | 0.482 |
| pcb1 | **0.74** | 0.679 | 0.672 | 0.672 | 0.603 | 0.602 | 0.496 |
| macaroni2 | **0.689** | 0.301 | 0.544 | 0.398 | 0.398 | 0.406 | 0.369 |
| macaroni1 | 0.627 | 0.464 | **0.709** | 0.409 | 0.627 | 0.62 | 0.455 |
| fryum | **0.588** | 0.534 | 0.573 | 0.519 | 0.481 | 0.473 | 0.389 |
| chewinggum | 0.687 | 0.699 | 0.675 | 0.711 | 0.747 | **0.762** | 0.735 |
| cashew | 0.664 | 0.664 | 0.557 | **0.733** | 0.534 | 0.52 | 0.656 |
| capsules | 0.429 | 0.371 | 0.371 | 0.41 | 0.371 | 0.369 | 0.4 |
| candle | 0.453 | **0.648** | 0.617 | 0.445 | 0.547 | 0.542 | 0.414 |
| mean | **0.631** | 0.592 | 0.585 | 0.568 | 0.558 | 0.554 | 0.537 |

|  | cfa | pfm | cflow | cdo | msflow | memseg | pefm | ast |
|---|---|---|---|---|---|---|---|---|
| pipe fryum | 0.79 | **0.839** | 0.8 | 0.806 | 0.653 | 0.681 | 0.831 | 0.56 |
| pcb4 | **0.722** | 0.531 | 0.536 | 0.696 | 0.541 | 0.564 | 0.562 | 0.662 |
| pcb3 | 0.496 | 0.512 | 0.304 | 0.472 | 0.288 | 0.402 | 0.432 | 0.171 |
| pcb2 | 0.489 | 0.474 | 0.35 | 0.438 | 0.394 | 0.435 | 0.511 | 0.504 |
| pcb1 | 0.58 | 0.611 | 0.589 | 0.618 | 0.527 | 0.501 | 0.534 | 0.12 |
| macaroni2 | 0.282 | 0.233 | 0.373 | 0.388 | 0.485 | 0.318 | 0.126 | 0.0577 |
| macaroni1 | 0.464 | 0.345 | 0.454 | 0.282 | 0.482 | 0.386 | 0.2 | 0.036 |
| fryum | 0.45 | 0.504 | 0.477 | 0.275 | 0.435 | 0.406 | 0.489 | 0.265 |
| chewinggum | 0.711 | 0.675 | 0.676 | 0.633 | 0.675 | 0.575 | 0.554 | 0.226 |
| cashew | 0.534 | 0.626 | 0.567 | 0.504 | 0.443 | 0.483 | 0.496 | 0.221 |
| capsules | 0.324 | 0.314 | 0.394 | 0.333 | 0.371 | 0.335 | 0.295 | **0.454** |
| candle | 0.555 | 0.461 | 0.52 | 0.508 | 0.469 | 0.443 | 0.438 | 0.368 |
| mean | 0.533 | 0.51 | 0.503 | 0.496 | 0.48 | 0.461 | 0.456 | 0.304 |

Table 9: VisionAD results on the VisA dataset based on the introduced PL metric.

25

For the interested reader we show the per class results and per dataset mean for each algorithm in Tables 2-9. These are what we believe to be the most comprehensive and fair set of results currently available for the MVTec and VisA datasets.

## Appendix C

Here we provide details of the subfunctions of PL introduced in Section 3.4.

### ExtractRotatedBoundingBoxes

Here we use the *minAreaRect* function from the OpenCV library (Bradski & Kaehler, 2020). This creates a rotated bounding box around every non-overlapping group of pixels.

### ScaleBBs

For every bounding box with a width and height of less than $1/8^{\text{th}}$ of the image width and height, the width and/or height is set to $1/8^{\text{th}}$. This is achieved easily because the bounding boxes are parameterised via center, width, height, and rotation.

### MergeOverlappingBBs

This is the most complicated of the sub functions. Each non-overlapping ground truth is compared with one another, and the overlap between the respective bounding boxes is calculated. Groups of bounding boxes are extracted, where for each group, one of the bounding boxes meets the overlap limit with another in the group. For each group, the underlying pixels used to create each respective bounding box are merged into one image, and a new rotated bounding box is created around the pixels in this image. Each group now represents one image and bounding box. The process is ran once again on the new images and bounding boxes. We find this process sufficiently merges close anomalies. We use an overlap limit of 0.33.

### CreateNearestBBFilters

For a group of bounding box, we wish to split an image into a set of masks, where each mask corresponds to a bounding box, and each mask highlights the pixels which are closest to its respective bounding box. This can be achieved by using the *cdist* function from the SciPy package (Virtanen et al., 2020). The distance between each pixel and bounding box center is calculated for each mask, and an *argmin* function is ran on the mask to get the one hot encoded final values. These filters are used to split the image into regions which correspond to each bounding box.

