# OpenReview forum: "VisionAD, a software package of performant anomaly detection algorithms, and Proportion Localised, an interpretable metric"
_TMLR — Accepted by TMLR_

### Review · Reviewer_xc77 · 2024-03-15

**Summary Of Contributions:**

In this paper, the authors propose VisionAD, a comprehensive library for anomaly detection in images designed for easy integration, experimentation, and fair benchmarking. It proposes standard API across all included algorithms to facilitate comparison, which improves the ease of integrating new algorithms.
They also introduce the Proportion Localised (PL) metric as a more intuitive and industrially relevant metric. This metric focuses on the proportion of anomalies that are adequately localized rather than pixel-perfect detection and aims to provide a more physically meaningful number for evaluation.

**Audience:**

Yes

**Claims And Evidence:**

Yes

**Requested Changes:**

Please address my concerns mentioned above.

**Strengths And Weaknesses:**

**Pros**

**Ease of Use and Integration:** VisionAD's design allows for the straightforward addition of new algorithms, encouraging rapid experimentation.
**Standardization:** The uniform API across algorithms aids in fair comparison and benchmarking.
**New Metric (PL):**  Offers an intuitive metric that addresses the limitations of existing anomaly detection metrics.

**Cons**

I have several concerns about the proposed metric, specifically the PL metric, which seems inadequate for the diverse conditions encountered in anomaly detection. Firstly, in situations where the ground truth is very thin or U-shaped, resembling a 45-degree angle—essentially, when the pixel-wise ground truth forms the diagonal of your bounding box—a perfect segmentation might not be recognized due to an Intersection over Union (IoU) lower than 0.3 with the bounding box. Secondly, in cases where the ground truth consists of small, non-continuous dots that a prediction fully covers, the PL metric inaccurately scores a 1 because it segments the prediction into many parts. This scenario indicates the model predicts a single anomaly, yet the PL score suggests perfect performance.

Conducting more experiments with the proposed metric on real-world datasets could be helpful in these issues.

---

> ### Author Response · Authors · 2024-03-21
> **Response to weaknesses and changes**
>
> Thank you for your comments. We are in the process of running more experiments. The concerns you have raised about the metric are valid.
> The first concern, regarding a long thin anomaly at 45 degree angle. You are correct in that this is an awkward shape for fitting a bounding box. The IoU limit of 0.3 was chosen because it allows mismatch between an anomaly shape and the bounding box. That being said, there could still be anomalies which are so awkwardly shaped that the 0.3 cannot easily be reached. We believe these are rare and the scenario you mentioned is an edge case. Our evidence for this happening rarely is that PL reports high scores across many datasets for a competent algorithm (Figure 2, Figure 5d). This would not be the case if the 0.3 was not possible to reach for a significant proportion of anomalies. The second column and fourth row of Figure 4 show a long thin anomaly at a diagonal angle, where the IoU limit is still reached. The concern you have mentioned is a weakness of the metric, and we are happy to add this in the limitations section. However, we believe that this edge case does not take away the positives of the metric, and the fact the metric mitigates many of the weakness of the traditional metrics P-AUC and AUPRO. It is the case that both the traditional metrics and the introduced metrics have pros and cons, and the user can choose the one they believe is best for their use case. We will add more experiments to justify our claim that this edge case is rare and does not take away from the positives of the metric.
> Your second concern is that there could be many close anomalies each a few pixels wide, which create many bounding boxes. You are correct this is undesirable behavior. Each bounding box would be scaled to have a width and height of 1/8th of the image, and all these anomalies would be counted as a separate anomaly, where they should be counted as one.
> We have not seen this happen, but you are correct that it could, so we are going to trial an edit to the PL calculation. If any scaled bounding boxes have >50% of their area in another bounding box, the smaller is merged into the larger, until no bounding boxes have 50% overlap with another. This would mean in the scenario of many close but tiny anomalies, they would all merge into one anomaly (scaled bounding box), as the previous scaled bounding boxes would all greatly overlap with each other.
> We are currently running the experiments on VisA, and hope to include another dataset as well, to justify the library and the metric.

---

> ### Author Response · Authors · 2024-04-02
> **CHANGES:**
>
> - We have updated PL to allow rotated bounding boxes (such that the metric is not dependent on image rotation). These rotated boxes are the smallest area boxes around a given anomaly. This means the PL score is independent of anomaly rotation, which solves your valid concern about a 45 degree anomaly. We have updated PL to merge closer bounding boxes. For a group of small and close anomalies, as their scaled bounding box labels would overlap, these anomalies are converted into one anomaly. New Figure 4 shows this clearly. This slightly reduces the number of bounding boxes, and slightly reduces the PL value. This effect is most apparent in the VisA dataset, where there are more small/close anomalies.
> - Thank you for this feedback we feel it has improved the metric.
> - We have included VisA dataset in the benchmarking and analysis of PL. This is a dataset of 12 classes across industrial and household objects, and is more challenging than MVTec.

---

### Review · Reviewer_Q98x · 2024-03-17

**Summary Of Contributions:**

This paper introduces the implementation of a new anomaly detection library. Specifically, the authors implement VisionAD, a new lightweight library for performance benchmarking. Then, the author proposes a new metric, Proportion Localized (PL), for measuring the performance of an anomaly detection algorithm.

The authors briefly describe the library's motivation and components. The library contains 15 different anomaly algorithms and several performance metrics to evaluate anomaly detection algorithms. This library would help develop a new anomaly detection method by efficiently comparing an algorithm to other existing algorithms.

Then, in Section 3, the author proposes a new performance metric included in the library. The proposed metric (PL) can overcome several disadvantages of the existing performance metrics: PL captures anomaly localization, has no noise from pixel-wise matching, and has clear physical meaning. With an experiment, PL’s performance measurement is compared to two other existing metrics. Section 4 covers more details about PL metric implementation with its detection result.

Section 5 presents the benchmark result for the anomaly detection algorithms in VisionAD on the MVTec dataset. In Section 6, the authors conclude the paper with discussions about limitations and potential future works.

**Audience:**

Yes

**Broader Impact Concerns:**

I don’t see a particular concern about the broader impact of this paper.

**Claims And Evidence:**

No

**Requested Changes:**

1. This paper should present more benchmarks with different configurations as a library paper. This will better demonstrate the library's usefulness.
   * Did the author try any dataset other than the MVTec dataset? Having similar results on a similar (but different) dataset would show the library's consistency in benchmarking.
   * Why did the authors keep the hyperparameters the same across each class? The library can easily support different hyperparameters, right?
   * How would this library support benchmarking those algorithms that change hyperparameters?
2. More quantitative experiments should be performed on the anomaly detection performance metric.
   * Figure 2 provides a good picture of PL's behavior, but we don’t know whether the observation is generalizable. An additional table showing the metric scores on different sets of data would be better.
   * Figure 4 can be a good way to show the authors’ choice of IoU limit, but it is better to provide a quantitative number that measures such a “satisfactory overlap” rather than spending a whole page with images.
3. Minor changes
   * I don’t think that the citation style follows the format in the [author guidelines](https://jmlr.org/tmlr/author-guide.html). Please use the [style file](https://github.com/JmlrOrg/tmlr-style-file/archive/refs/heads/main.zip) provided in the author guidelines.
   * Please proofread the paper once again and fix all the spelling and grammar errors.

**Strengths And Weaknesses:**

## Strengths
1. The paper is well-motivated, and the library seems valuable for benchmarking purposes.
2. The library contains many anomaly detection algorithms.
3. From the visualization results, the performance of the proposed metric seems reasonable.

## Weaknesses
1. Having two different contributions simultaneously is good, but I don’t see enough support for each contribution. See **Requested Changes** for more details. To be honest, I don’t see the reason the authors present both VisionAD and PL with a short submission limit. I’d suggest adding more experimental support for each and submitting them as long submissions.
2. The paper contains several minor problems, e.g., citation style, typos, and grammar.

---

> ### Author Response · Authors · 2024-03-21
> **Response to weaknesses and changes**
>
> Thank you for the comments. We struggle to see how we could present the ideas separately.  This is because the new metric is linked to the library, and the results for the new metric are calculated using the library. We are in the process of running experiments on other datasets. We are also proof reading and fixing typos and grammar.
> We submitted as a short format as we want to keep the main message of the paper concise, i.e. a lightweight but flexible library, and an intuitive metric. We believe the two concepts can be presented concisely, and this will encourage engagement/use/reading. We are happy to add more experiments/justification at the end of the paper or in the appendix. Does this approach sound satisfactory?
> We are in the process of running the VisA dataset, and we will add the results.
> The library does allow hyperparameter changes in an intuitive manner. However, we use the same hyperparameters because we believe this is the fairest method for benchmarking. Using specific hyperparameters for specific classes of the MVTec dataset could be considered test set overfitting, as one could sweep across the hyperparameters to find the best for each class. In practice, when given a novel problem, one does not know the best hyperparameters for that problem. Therefore we use the hyperparameters presented in the original publications across all classes. Do you agree that using the same hyperparameters for a given algorithm across each class is fairest for benchmarking?
> You have raised a good point in that the library is not currently best equipped for changing hyperparameters for a given dataset. We have added logic where if the model dictionary contains a dataset key, it will default to that dataset, and ignore the dataset list. This means a user can set different hyperparameters for a given dataset (your mentioned use case) by including a dataset key in the model dictionary, but can also perform a sweep over datasets using one hyperparameter set (by not including a key in the given dataset - here the code will default to the dataset list presented in the config). Adding this option does increase complexity slightly, but we believe it is justified to enable the use case of different hyperparameters for different datasets.
> Figure 2 was intended to be very general. I can replicate Figure 2 but for different sets of data. Maybe a scatter plot showing different combinations of algorithms/classes, and their scores for each metric? This would provide more evidence towards the behavior of the metric.
> I will fix the citation format and proofread again.

---

> > ### Author Response · Authors · 2024-03-21
> >
> > Regarding your comment of the 0.3 IoU limit. The community can choose whichever limit they feel is best. We recommend 0.3 because from visual inspection that is when we believe the anomalies are located sufficiently. We provide a page of images so readers can also make their own visual inspection and come to their own conclusion. Different use cases may require different IoU limits, however for benchmarking, we feel 0.3 is the best value to use. Unfortunately we don’t see a way of deriving this number mathematically. Sometimes the case exists that numbers have to be used which are not derived mathematically. The AUPRO calculation uses 0.3 where this 0.3 has no mathematical derivation. Is you have any advice regarding this please let us know.

---

> > ### Author Response · Authors · 2024-04-02
> > **CHANGES:**
> >
> > - We have included VisA dataset in the benchmarking and analysis of PL. This is a dataset of 12 classes across industrial and household objects, and is more challenging than MVTec
> > - We have generalised Figure 2 to compare the metric's behaviour using the real results, as opposed to a synthetic experiment. I believe this Figure clearly shows the superior behaviour of PL, using real results.
> > - We use the same hyperparameters across each class for fairness, to avoid over tuning of a hyperparameter set to a dataset (test set leakage/overfitting)
> > - However, the wrapper does allow hyperparameter changing for a given dataset
> > - For Figure 4, we have updated the Figure to make it more clear, and we take precedence for the 0.3 limit from the use of 0.3 in the AUPRO metric from anomaly detection. The community is free to choose another IoU limit is they feel another is more suitable
> > - We have fixed the citation format and proof read
> >
> > - We have updated PL to allow rotated bounding boxes (such that the metric is not dependent on image rotation), and we have updated PL to merge closer bounding boxes, which slightly reduces the number of bounding boxes, and slightly reduces the PL value. This effect is most apparent in the VisA dataset, where there are more small/close anomalies
> >
> > Thank you for your feedback

---

### Review · Reviewer_rj9M · 2024-03-19

**Summary Of Contributions:**

This paper proposes a new software package for the vision anomaly detection algorithms and hosts a leaderboard such that the researchers could fairly compare among different algorithms; moreover, this paper proposes a new metric, which provides more straight-forward interpretation.

**Audience:**

Yes

**Claims And Evidence:**

No

**Requested Changes:**

See Weaknesses

Here are other comments

1. In section 1.1, when comparing to Anomalib, the authors claim the visionAD is more light-weight to use but it is unclear to me, if VisionAD become larger and larger, wouldn't it have the similar issue? Will it become another anamalib, which is comprehensive but more efforts? Can the authors elaborate more?

2. how flexible of the proposed VisionAD to adopt algorithms added into Anomalib? So, the researchers do not need to duplicate their work to different frameworks.

3. As the proposed PL needs the bounding box label; what if the users incorporate that into P-AUC and AUPRO calculation? Does that solve the weakness the authors mentioned?

4. Caption for figure 3, there is no color red used in the figure.

5. In figure 5, under different metrics, the rank of algorithms are different; how should the users to really compare the algorithms?

**Strengths And Weaknesses:**

Strengths

1. As the ML research progresses rapidly, it is good to have a platform to compare results fairly.
2. The proposed software architecture is easy to integrate any new AD algorithms.

Weaknesses:

1. As this task is closer to semantics segmentation or instance segmentation, the Dice score should be compared.
2. The difference from the normal metrics used in object detection is unclear; moreover, in object detection, an average score across different IoU is reported.

---

> ### Author Response · Authors · 2024-03-21
> **Response to weaknesses and changes**
>
> You are correct in that the introduced metric is closer to instance segmentation. We had the choice between IoU and Dice Score in the introduced metric. We chose IoU because we believe it is more intuitive/explainable to a non-data scientist than dice score. We can add a paragraph explaining this decision if you wish? If we added dice score in the results it may confuse the reader, and we would also need to choose a prediction threshold to calculate dice score.
> There is a major difference between the metrics used in object detection in that they are not directly applicable. In anomaly detection the algorithms output heatmaps, whereas in object detection the algorithms output bounding boxes (YOLO/R-CNN). Therefore one could not directly apply the metrics from object detection (mAP50, mAP25), to anomaly detection. This is the reason for the novel algorithm in Algorithm 1. We could have calculated an average across IoU for the PL calculation, but I believe it is more useful to an industry profession to know how many anomalies have been located to a sufficient level. This is why we choose to report proportion of anomalies with IoU>0.3.
> The library leaves out certain features that Anomalib has, such as edge inference. Algorithms can be added in by filling out the boilerplate code shown at the start of Appendix A. We already have a larger quantity and more recent algorithms than Anomalib. Despite having less algorithms, we believe it to still be far more lightweight; it can be installed by just running git clone and installing the packages required for the desired algorithms. We don’t have a set of different installation options like Anomalib.
> Algorithms can be easily added by filling in the methods listed at the start of Appendix A. We don’t make a special case for directly transferring algorithms from Anomalib to VisionAD. We believe this would add complexity, we want all algorithms to follow the same API. If one knows the algorithms well they would be able to quickly take the Anomalib code and refactor it to fit the VisionAD API.
> One could create the bounding box labels, then run the P-AUC and AUPRO, this would fix some of the weaknesses such as noise from exact pixel matching, but it would not give an intuitive number, and it would not report the number of anomalies located. These would still have the weakness of not reporting values linearly between 0 and 1 (see Figure 2).
> One should compare the algorithms via the metric which aligns to their goals. If they are looking for pixel level precision, and they are looking for the most area under the ROC curve (i.e. best pixel separation across many different thresholds), then P-AUC is best. P-AUPRO weighs each anomaly equally, but also gives an unintuitive number. These metrics has mathematical precision, but we believe they miss the point of reporting which algorithm performs the task the best for a real world purpose, and they report an unintuitive number. As mentioned above, these metrics do not report linearly between 0 and 1, P-AUC tends to report very high numbers 0.98+, whilst P-AUC tends to report ~0.92 for a very strong algorithm. Finally these algorithms will be taking into account noise from the process of matching exact pixels. The labeller would have labelled ambiguous pixels near the boundaries of anomalies or inside the curvature of an anomaly as anomalous or regular, and this would be taken into account by these metrics.
> Apologies for the confusion in Figure 3. In the final 5 rows, the red smudge demonstrates the predicted pixels from the algorithm (the heatmap put through the given threshold). This figure is purely for demonstrating the splitting of an image for different anomalies. I will make it more clear.
> I propose that we are more interested in the proportion of anomalies that located by the model (where located refers to meeting the 0.3 IoU), therefore I would recommend PL, This mitigates the weaknesses above by giving an initiative value that linearly moves between 0 and 1 as algorithm competence increases, and mitigates the issue of the labels of ambiguous pixels through the 0. 3 IoU threshold. The user is free to set another IoU threshold if they wish, but we present 0.3 as we believe it is the most suitable.

---

> > ### Author Response · Authors · 2024-03-21
> >
> > I didn't mention earlier, thank you for your comments.

---

> > > ### Author Response · Authors · 2024-04-02
> > > **CHANGES:**
> > >
> > > - We have rarely seen dice score used in anomaly detection. We choose IoU to use in PL because we believe this is more interpretable, and one of the main aims of this metric is interpretability. To report dice score we would to choose a threshold to convert the continuous prediction into a binary prediction. Dice score does not provide a proportion of anomalies found, weights each anomaly differently depending of size, is lesser known outside of computer vision experts, therefore we do not believe it is a direct competitor of PL, and therefore we believe that comparing to dice score would be unnecessary and may introduce unclarity/confusion.
> > > - We believe users should rank the algorithms via PL, due to its advantages of: interpretability, functioning at the anomaly level, reporting across the greatest range, and negating any noise from ambiguous pixel labelling.
> > > - The introduced metric PL is very different from other object detection metrics. The object detection metrics compare a bounding box prediction to a bounding box label (YOLO/R-CNN). However PL compares the anomaly detection heatmap prediction, to the rotated bounding box label, and provides a system for automatically creating these labels, and counting the number of anomalies sufficiently found. One component of PL is IoU, like IoU is a component of the metrics from object detection (mAP/AP). However PL is significantly different from these metrics.
> > > - Putting bounding boxes AUPRO and PAUC would only solve the issue of ambiguity in pixelwise labels. It would not solve the other disadvantages of no interpretability, predicting over a small range, and the metrics working at the pixel level, not anomaly level
> > > - To create an algorithm in VisionAD, a user only needs to fill in 5 methods. Anomalib requires far more such as ONNX compatibility. VisionAD and Anomalib completement each other in different ways.
> > > - We do not have a precedence for converting Anomalib code to VisionAD, however someone familiar with a given algorithm would be able to add it to VisionAD very fast.

---

### Decision · Action_Editor_1zmD · 2024-05-01

**Recommendation:** Accept with minor revision

**Comment:**

The authors propose a new software package for the vision anomaly detection algorithms and hosts a leaderboard such that the researchers could fairly compare among different algorithms; moreover, this paper proposes a new metric, which provides more straight-forward interpretation.

The paper is well-motivated, and the library seems valuable for benchmarking purposes. The library contains many anomaly detection algorithms. From the visualization results, the performance of the proposed metric seems reasonable. However, the majority part of the work is about the software, and the paper does not provide well-supported claims and evidence for each contribution, and I believe the authors should provide more results. Therefore, based on three qualified reviews, this paper can be accepted with minor revision and the authors are encouraged to merge the comments into their update versions.

**Audience:**

Yes

**Claims And Evidence:**

Yes